# Fate of antibiotic resistant *E. coli* and antibiotic resistance genes during full scale conventional and advanced anaerobic digestion of sewage sludge

Sky Redhead[1], Jeroen Nieuwland[1], Sandra Esteves[1], Do-Hoon Lee[2], Dae-Wi Kim[2], Jordan Mathias[3], Chang-Jun Cha[2], Mark Toleman[3], Richard Dinsdale[1], Alan Guwy[1], Emma Hayhurst[1]*

1 Faculty of Computing, Engineering and Science, University of South Wales, Pontypridd, United Kingdom, 2 Department of Systems Biotechnology and Centre for Antibiotic Resistome, College of Biotechnology & Natural Resources, Chung-Ang University, Anseong, Republic of Korea, 3 School of Medicine, Cardiff University, Cardiff, United Kingdom

* emma.hayhurst@southwales.ac.uk

**Data Availability Statement:** All relevant data are within the paper and its Supporting information files.

## Abstract

Antibiotic resistant bacteria (ARB) and their genes (ARGs) have become recognised as significant emerging environmental pollutants. ARB and ARGs in sewage sludge can be transmitted back to humans via the food chain when sludge is recycled to agricultural land, making sludge treatment key to control the release of ARB and ARGs to the environment. This study investigated the fate of antibiotic resistant *Escherichia coli* and a large set of antibiotic resistance genes (ARGs) during full scale anaerobic digestion (AD) of sewage sludge at two U.K. wastewater treatment plants and evaluated the impact of thermal hydrolysis (TH) pre-treatment on their abundance and diversity. Absolute abundance of 13 ARGs and the Class I integron gene *intI1* was calculated using single gene quantitative (q) PCR. High through-put qPCR analysis was also used to determine the relative abundance of 370 ARGs and mobile genetic elements (MGEs). Results revealed that TH reduced the absolute abundance of all ARGs tested and *intI1 by 10–12,000 fold*. After subsequent AD, a rebound effect was seen in many ARGs. The fate of ARGs during AD without pre-treatment was variable. Relative abundance of most ARGs and MGEs decreased or fluctuated, with the exception of macrolide resistance genes, which were enriched at both plants, and tetracycline and glycopeptide resistance genes which were enriched in the plant employing TH. Diversity of ARGs and MGEs decreased in both plants during sludge treatment. Principal coordinates analysis revealed that ARGs are clearly distinguished according to treatment step, whereas MGEs in digested sludge cluster according to site. This study provides a comprehensive within-digestor analysis of the fate of ARGs, MGEs and antibiotic resistant *E. coli* and highlights the effectiveness of AD, particularly when TH is used as a pre-treatment, at reducing the abundance of most ARGs and MGEs in sludgeand preventing their release into the environment.

**Funding:** EH Ser Cymru NRN LCEE Returning Fellowship http://www.nrn-lcee.ac.uk/returning-fellowships.php.en SE, EH, SR Welsh Government SMART Expertise https://businesswales.gov.wales/expertisewales/support-and-funding-businesses/smart-expertise The funders had no role in study design, data collection and analysis, decision to publish, or preparation of the manuscript.

**Competing interests:** The authors have declared that no competing interests exist.

# Introduction

Under a 'One Health' approach, the environment is increasingly being recognised as an essential part of the transmission cycle for antibiotic resistant bacteria (ARB) and their genes (ARGs) [1]. In recent years, ARBs and ARGs have become recognised as significant emerging environmental pollutants [2]. ARB and ARGs in the environment can find their way back to humans via direct or indirect contact with contaminated water or land, via drinking water abstracted from contaminated rivers, or via the food chain [3].

Wastewater treatment plants (WWTPs) are seen as key potential gateways for controlling the dissemination and release of ARB and ARGs into the environment [4]. WWTPs bring together vast amounts of waste from a variety of different sources [5], with the resulting accumulation of extremely large numbers of bacteria, a significant quantity of which come from the human gastrointestinal tract. Some of these bacteria will carry resistance genes either in their core genome or on mobile genetic elements (MGEs), and, particularly under selection pressure, these can be transferred from one bacterium to another via horizontal gene transfer [6, 7].

Previous studies have documented the effectiveness of WWTPs at removing ARB and ARGs from liquid effluent released to the aquatic environment [8–10], although in some WWTPs certain ARB and ARGs appear to be enriched [11–13]. However, it is estimated that 90–95% of ARB and ARGs in WWTPs are associated with the sludge fraction [14]. This makes sewage sludge treatment a key process for preventing ARB/ARG release into the environment. Sewage sludges are typically separated from the liquid fraction during primary and secondary settling, and then potentially combined for further treatment. Depending on the country, sewage sludges can either be incinerated, put to landfill, or treated prior to recycling to agricultural land as a fertiliser/soil conditioner. A common method of sewage sludge treatment is anaerobic digestion (AD), which is effective at stabilising sludge, producing renewable energy, and reducing the pathogen load to levels appropriate for reuse [15]. Post digestion, biosolids are often applied to agricultural land as a soil conditioner, returning valuable nutrients in the form of N, P, and K, as well as organic matter to the soil. However, the reuse of sewage sludge also poses a potential risk of direct transmission of ARB and ARGs back into humans via the food chain, or indirectly via agricultural run-off into aquatic ecosystems [3]. Sewage sludge also contains a number of other emerging contaminants such as antibiotics and other organic contaminants, heavy metals and microplastics which may also act as selection pressures for antibiotic resistance, exacerbating their spread [16].

In contrast to treatment of the aqueous component of sewage, there is a limited number of studies that examine the fate of ARB and ARGs in full-scale sewage sludge treatment. Of those that do, many of them only focus on a small number of ARB and ARGs. The inefficient removal of these potential contaminants is a common finding in these studies [e.g. 8, 17]. The loading rate of ARGs into the environment from sludge is estimated to be 1000 times than from aqueous sewage effluent [18], making sludge management a critical step in the control of environmental ARG release [19]. With over 13 million tonnes of sludge applied to agricultural land each year in the European Union alone, understanding and reducing the potential public health risks associated with sludge recycling to land is recognised as an urgent research gap [19]. There is very little evidence available to characterise the fate of ARB and ARGS once sludge is applied to land, or the relative importance of this particular environmental transmission route for public health. Until there is a better understanding, quantifying ARB and ARGs in particular environments (including sludge) has been suggested as a proxy for risk [20].

The high prevalence of ARGs in sludge combined with the relative inefficiency of AD for ARG removal makes characterising the removal efficiency of various modifications to AD a

research priority in order to inform industry on the best way to reduce AMR-associated risk of sludge recycling.

Prior to AD, sludge can be pre-treated in a number of ways. Advanced digestion using thermal hydrolysis (TH) as a pre-treatment is increasing in deployment [e.g. 20, 21]. The sludge during the TH process undergoes a temperature and pressure increase followed by a rapid pressure release. This has been found to improve methane yields from digestion [22], to allow an increase in sludge throughput and to more effectively remove pathogens [23, 24]. Despite its growing popularity, only one previous study has investigated the fate of ARGs during full-scale TH-AD with analysis of only 14 ARGs [25]. In addition, culture-dependent methods can provide essential information on direct risk of ARB transmission into the environment, but full-scale studies on the fate of ARB during AD have been rare. Therefore, this study was undertaken with the following objectives:

1. To evaluate the impact of thermal hydrolysis pre-treatment of sludge on ARB and ARGs during AD

2. To investigate the fate of ARB and ARGs in sewage sludge during AD

To overcome knowledge gaps of other studies, the relative abundance of 370 ARGs and mobile genetic elements (MGEs) and the absolute abundance of 13 ARGs (from a range of antibiotic classes) and *intI1* was quantified from two WWTPs using mesophilic AD (MAD) to treat sludge. One of the plants used TH as a pre-treatment, the other did not. Samples were taken before and after TH and after subsequent MAD at the first plant, and before and after MAD at the second plant. The number of *Escherichia coli* colony forming units (CFUs) from each sampling point was also quantified and the antibiotic susceptibility profiles of 300 *E. coli* isolates to nine antibiotics was characterised. All *E. coli* isolates were classified into phylotypes to determine whether sludge treatment resulted in a shift in the *E. coli* community.

## Materials and methods

### Study location and sample collection

Samples were collected from two centralised WWTPs in the UK both treating sewage via an aerobic treatment stage which generates an indigenous secondary sludge. Both plants also import secondary sludges as well as primary sludges from satellite sites. Both plants operate on a mixture of primary and secondary sludges, which then undergoes an anaerobic treatment. WWTP1 employs advanced AD, using TH prior to mesophilic AD (TH-MAD). At this WWTP, samples (approx. five litres) were taken of thickened sludge prior to pre-treatment, after sludge had been thermally hydrolysed (165°C for 30 minutes), and after MAD. From WWTP2 which employs conventional MAD to treat sludge with no pre-treatment, samples were taken of thickened sludge before MAD and after MAD. Fig 1 shows the details of the treatment processes and sample points at each WWTP. WWTP1 treats primary and secondary sludge from its own treatment process (which uses an aerobic sequencing batch reactor SBR) (70% by volume), and also imports sludges from other WWTPs (30% by volume). Mixed sludge is thickened through centrifuges to approximately 16% dry solids (DS) content and thermally hydrolysed. TH sludge is then diluted to approximately 8–9% DS with filtered and UV disinfected final effluent from the SBRs and anaerobically digested (digester retention times vary typically between 12–25 days) at mesophilic temperature (typically 37–42°C) before a short aerobically sparged storage. From here the digestate follows to belt presses for dewatering and whilst the digestate liquors return to the treatment process, the solid digestate (cake) is transported to agricultural land. WWTP2 treats indigenous primary and secondary sludge

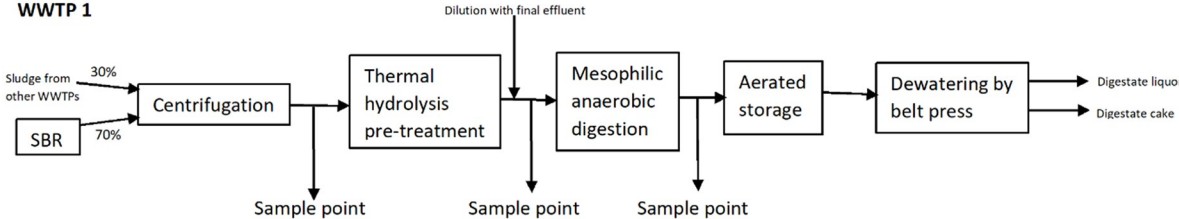

**Fig 1. Showing the sludge treatment processes and sampling points for each WWTP.** SBR = sequencing batch reactor. Final SBR effluent post filtration (300 μm) is used for dilution of the sludge post TH whilst TH sludge stream is still hot (>80˚C).

from activated sludge treatment. Sludge is thickened to approximately 5–7% DS before being anaerobically digested (retention time varied between 17 and 22 days in the sampling period) at mesophilic temperatures (32.6–38˚C over the sampling period). The digestate is then centrifuged, with the liquor returned to the treatment process and the digestate cake transported to agricultural land. In total, four sets of samples were collected from WWTP1, and five sets of samples were collected from WWTP2, between early Spring and mid-Autumn (span of 9 months).

## Processing of samples for enumeration of *E. coli* and total heterotrophs

All samples were processed within four hours of collection. To ensure samples were representative, each sample (approx. five litres) was thoroughly stirred to resuspend settled sludge. 20 mL of the stirred sludge was then taken and saved for further analysis. A serial dilution of each sample was prepared in triplicate and plated onto Membrane Lactose Glucuronidase Agar (MLGA) (Oxoid) for isolation of *E. coli*, and nutrient agar for the enumeration of total heterotrophs. MLGA plates were incubated at 37˚C for 24 hours, and nutrient agar plates were incubated at 30˚C for 48 hours. Colony forming units (CFUs) of *E. coli* and total heterotrophs per gram of dry weight (g$^{-1}$ dry wt) were calculated.

## Antibiotic susceptibility testing of *E. coli*

A set of 300 isolated colonies of *E. coli* were randomly selected for antibiotic resistance profiling against clinically relevant antibiotics. These included 100 from untreated sludge from each WWTP (sample points 1 and 4) and 100 from after MAD at WWTP two (sample point 5). All antibiotic testing was carried out according to CLSI disk diffusion guidelines (Clinical Laboratory Standards Institute, 2017). Selected isolates were first re-streaked from MLGA onto nutrient agar and incubated for 24 hours at 37˚C. The isolate was then resuspended in phosphate buffered saline (PBS) to an OD$_{600}$ of 0.5 using a MacFarland standard, plated onto Müller-

Hinton agar (Oxoid) and incubated at 35°C for 18–20 hours. Zones of inhibition were measured to the nearest millimetre.

The antibiotics used were ciprofloxacin (5 μg), gentamicin (10 μg), meropenem, (10 μg), ampicillin (10 μg) (used to test for amoxicillin resistance (Clinical Laboratory Standards Institute, 2017)), trimethoprim (5 μg), and four different third generation cephalosporins—ceftazidime (30 μg), cefoperazone (75 μg), ceftiofur (30 μg) and cefpodoxine (30 μg) (all antibiotics were from Oxoid., Basingstoke, UK).

## Phylotyping of *E. coli* isolates

Phylotyping of the 300 *E. coli* isolates was carried out according to the Clermont method [26]. Briefly, a quadruplex PCR reaction was set up for the *E. coli* genes/genome regions *chuA*, *yjaA*, TspE4.C2, *arpA*. The resulting banding pattern allows the determination of the specific *E. coli* phylotype. To distinguish between phylotype A and C or D and E, a further confirmatory PCR was carried out using primers specific for *trpA* or *arpA*, respectively.

## Quantification of ARGs and *intI1* using single gene and high-throughput quantitative PCR

Two methods of quantitative (q) PCR were used in this study. Single gene qPCR gives absolute abundance measures which are important for quantitative environmental monitoring [27], and it is a more sensitive technique. High throughput qPCR provides a more comprehensive picture of the fate of ARGs within the sludge treatment plant and does not limit researchers to pre-selected genes. However, it is generally a less sensitive technique than single gene qPCR and only relative abundance can be quantified.

DNA was extracted from samples of centrifuged sludge (250 milligram) using the QIAGEN DNeasy PowerSoil Kit (QIAGEN, Manchester, UK). Extracted DNA was stored at -20°C until further use. The quantity of extracted DNA was measured using Qubit, and extractions were considered good enough quality for downstream analysis with a Qubit reading of 30 ng/μL or more.

For single gene qPCR, the absolute abundance of 14 different genes were analysed using a SYBR Green (QIAGEN) qPCR assay. These genes conferred resistance to a range of different antibiotic families including the tetracylines (*tetM*), the β-lactamases (bla-CTX-M families 1 and 9 and *blaIMP*), the fluoroquinolones (*qnrS*), the aminoglycosides (*aac(3)-1*), trimethoprim (*dfrA1*, *dfrA5*, *dfrA7* and *dfrA12*), MLSB antibiotics (*ermF*) and sulphonamides (*sul1*). The class one integron gene (*intl1*) and the 16S rRNA gene were also analysed. Primers, annealing temperatures and the reaction matrix are described in S1 and S2 Tables.

A rotogene series Q (QIAGEN) was used for the qPCR assay. Samples were processed in triplicate. Each run consisted of an initial denaturation step of 94 °C for 5 minutes followed by 35–60 cycles of 95 °C for 10 seconds then 60 °C for 10 seconds. Total amounts of the gene of interest were calculated from a standard curve created using dilutions of a small section of the gene (idtDNA, Leuven, Belgium), this was used to calculate copy number (g$^{-1}$ dry wt) of sludge sample for absolute abundance. Melt curves were analysed for quality control, and selected PCR products were sequenced to ensure specific amplification of the target gene.

For high throughput qPCR, ARGs and MGEs in the samples were detected and quantified using SmartChip Real-time PCR system (Takara, Japan), a high-throughput quantitative PCR system. Primer set 2.0 consisting of 312 ARG-targeted and 58 MGE-targeted primers was employed for the analysis [28]. PCR reaction, data collection, CT value calculation, normalization and abundance calculation of ARGs and MGEs were conducted as previously described [29, 30].

## Data analysis

Statistical analysis of data was carried out using SPSS version 23. Chi squared goodness of fit analysis was used to determine any significant differences between the percentages of resistant *E. coli* in each sample. A Mann Whitney U test was used to determine whether any of the observed differences in absolute abundance of resistance genes between samples was significant. Significance was determined at a P>0.05. The composition and abundance of ARGs and MGEs at each sample were used for the principal coordinates analysis (PCoA) based on Bray-Curtis dissimilarity matrix. The analysis and plotting were conducted using a PAST program [31].

## Results

### Fate of antibiotic resistant *E. coli* and total heterotrophs during MAD and TH-MAD

*E. coli* CFUs from WWTP1 ranged from 3.8 x $10^5$ to 5.7 x $10^5$ (mean 5.1 x $10^5$ ± 7.1 x $10^3$ SD) $g^{-1}$ dry wt in untreated sludge. After TH there was no detectable growth of *E. coli* on MLGA (i.e. less than 1 CFU per gram), and there was no resurgence of *E. coli* detected after subsequent MAD. *E. coli* CFUs from WWTP 2 ranged from 4.6 x $10^5$–8.3 x $10^5$ (mean 5.8 x $10^5$ ± 1.1 x $10^5$ SD) CFUs $g^{-1}$ dry wt of untreated sludge. After MAD, the number of detectable *E. coli* CFUs on MLGA was between 2.7 x $10^3$–9.6 x $10^3$ (mean 3.7 x $10^3$ ± 2.9 x $10^3$ SD) CFUs per gram dry weight, a decrease of more than 99% (Fig 2A).

Resistance to amoxicillin, trimethoprim, ciprofloxacin and gentamicin was found in *E. coli* isolates in untreated sludge at WWTP1, and before and after MAD at WWTP2 (Fig 2B). Resistance to third generation cephalosporins was also found before and after MAD at WWTP2 but was not detected in WWTP1. Resistance to meropenem was not detected at either WWTP throughout this study. Prevalence of amoxicillin resistance was highest, with 42% and 46% of isolates from untreated sludge at WWTP1 and WWTP2 showing resistance, respectively. Trimethoprim resistance increased significantly from 7% to 16% during MAD at WWTP2 (P = 0.046).

One multi-drug resistant (MDR) isolate (resistant to three or more antibiotic families) was found in untreated sludge and two MDR isolates were found after MAD at WWTP2. Four MDR isolates were found in untreated sludge from WWTP1.

The pattern of resistance found in both WWTPs combined is comparable to the pattern of clinical levels of resistance in Wales, with the percentage of *E. coli* isolates from WWTPs resistant to each antibiotic tracking between 8.3 and 29% below the percentage of *E. coli* isolates from the clinic resistant to each antibiotic [32] (S1 Fig).

The phylotype composition of *E. coli* isolates from before and after MAD at WWTP2 was compared to determine whether there was any shift in phylotypes during AD. Phylotype A and B2 were the most dominant phylotypes (each 24% before MAD and 30% (A) and 22% (B2) after MAD), and no significant changes were observed (S2 Fig).

### Fate of ARGS and MGEs during TH-MAD and MAD

Fig 3 shows the absolute abundance of ARGs in sewage sludge. All 14 genes tested for were detected in untreated sludge samples from both WWTPs. The most abundant ARG in untreated sludge was *sul1* (1.9 x $10^7$–1.3x$10^8$ copies $g^{-1}$ dry wt). After thermal hydrolysis at WWTP 1, there was a significant reduction in absolute abundance of all genes tested. The largest reduction was in *bla-Imp*, which showed greater than a 12,000 fold decrease. The smallest reductions were seen in CTX-M-9 which exhibited less than a 10 fold decrease (Fig 3A).

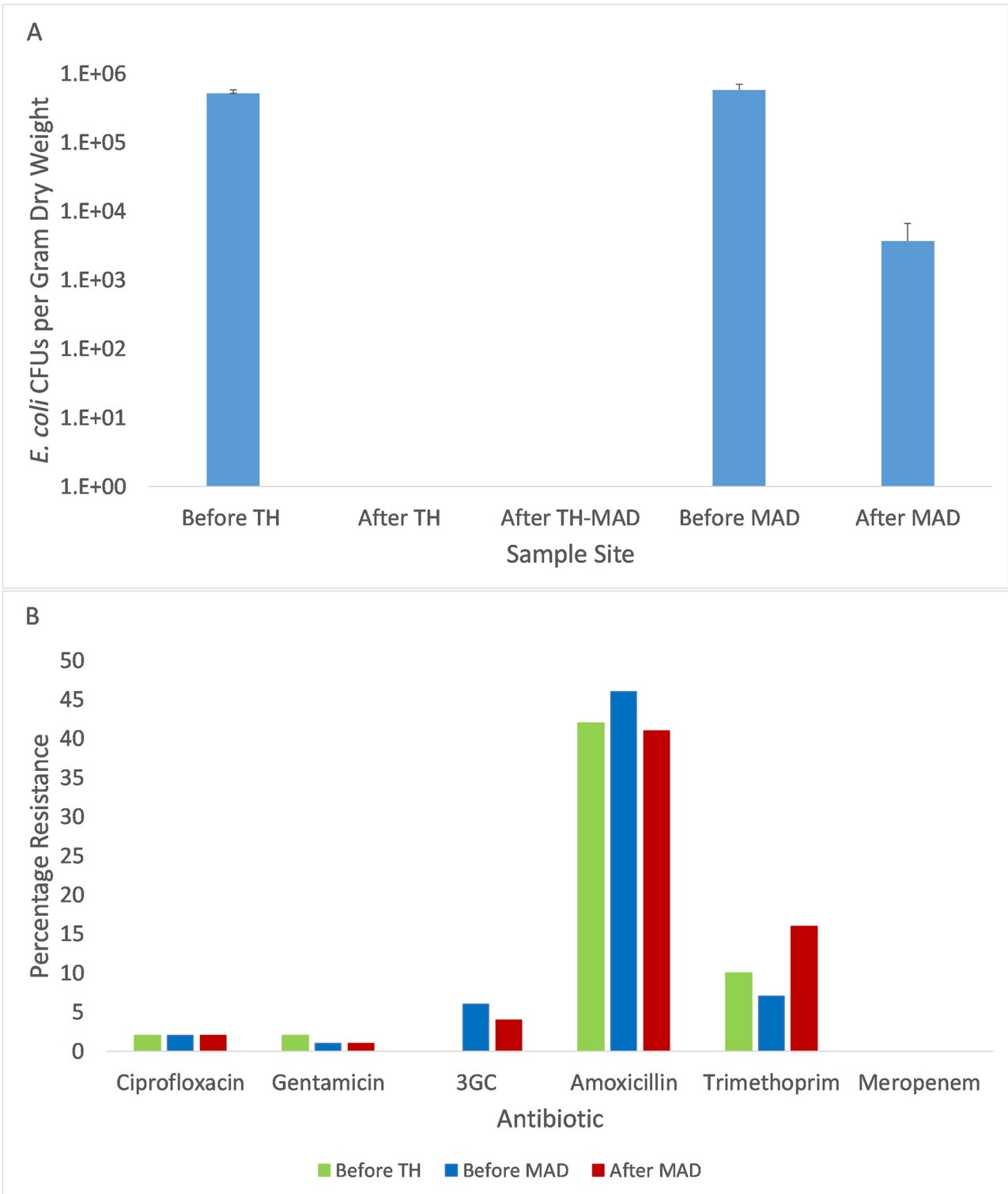

**Fig 2. A)** *E. coli* CFUs in sludge from WWTP 1 and WWTP2. This shows that there was no detectable *E. coli* after thermal hydrolysis at WWTP 1, and no resurgence in detectable *E. coli* numbers after subsequent MAD. MAD achieved a 2–3 log10 reduction in *E. coli* CFUs. **B)** Prevalence of antibiotic resistance in *E. coli* isolated from sewage sludge. Graph shows *E. coli* resistance rates to six antibiotics at WWTP 1 before TH-MAD, and at WWTP 2 before and after MAD. The four third generation cephalosporins (3GCs) tested have been grouped together. No *E. coli* were tested for resistance after TH or after TH-MAD because no *E. coli* were detected.

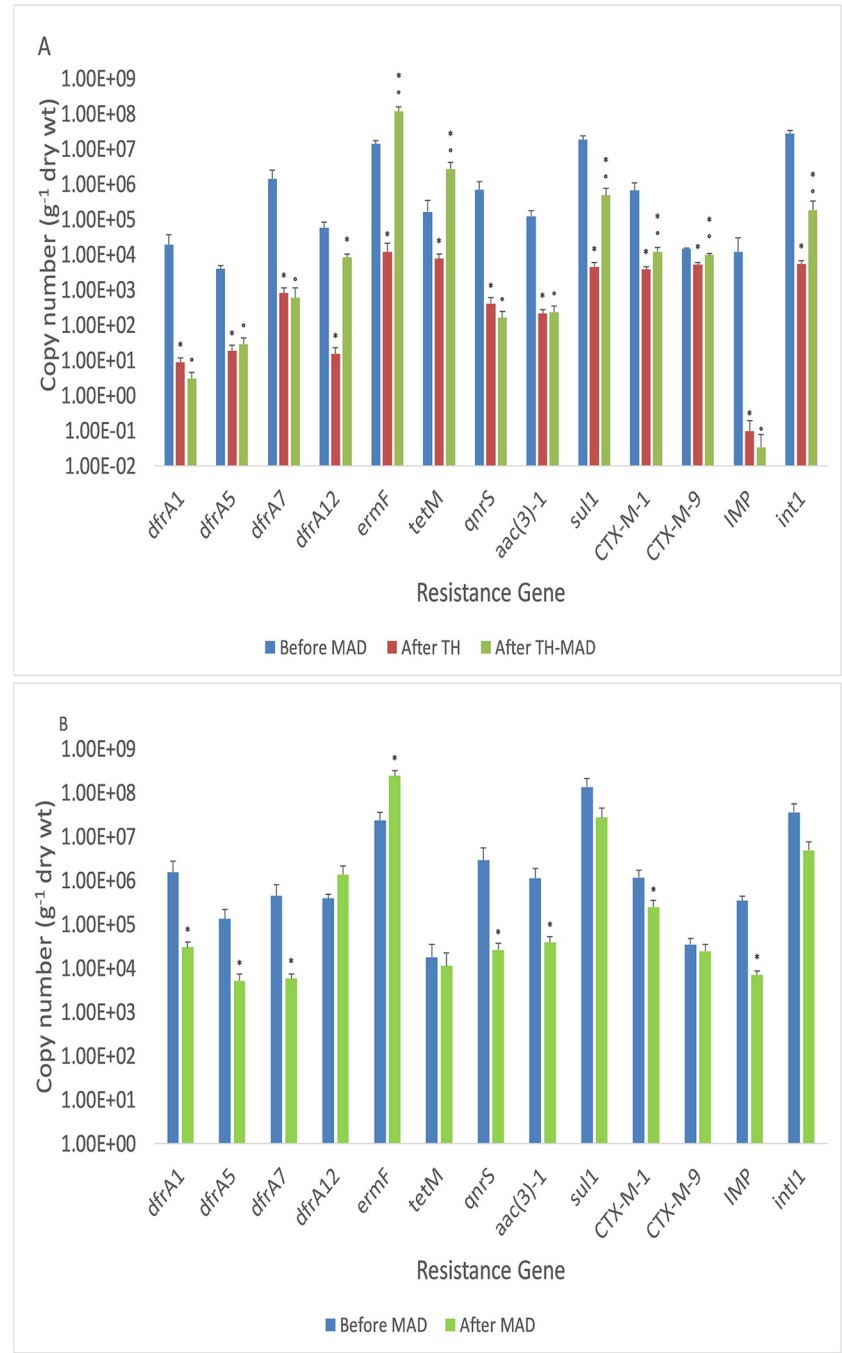

**Fig 3. Absolute abundance of ARGs in sewage sludge. A)** Absolute abundance (copy number g⁻¹ dry wt) of all 13 resistance genes tested, and *intI1*, during sludge treatment at WWTP 1. Statistically significant changes between samples is indicated by a * and statistically significant changes between feed sludge and digestate is indicated by a ˚. **B)** Absolute abundance (copy number g⁻¹ dry wt) of all 13 resistance genes tested, and *intI1* during sludge treatment at WWTP 2. Statistically significant changes from before to after MAD are indicated by a *.

The absolute abundance of many genes then rebounded after MAD at WWTP1 (Fig 3A). Significant increases during MAD were observed in *ermF*, CTX-M-9, CTX-M-1, *dfrA12*, *tetM*, *sul1* and *intI1*. No resistance gene tested showed a significant decrease. The biggest increase was seen in *ermF*, from $1.2 \times 10^4$ to $1.2 \times 10^8$ copies (g$^{-1}$ dry wt). In the case of most genes, the rebound during MAD was not sufficient to increase abundance beyond the level of untreated sludge, meaning that an overall reduction in resistance gene load was seen during treatment (between samples 1 and 3). The exceptions were *ermF* and *tetM* which both increased about 10-fold from before to after TH-MAD.

After MAD at WWTP2, the absolute abundance of many genes (*dfrA1*, *dfrA5*, *dfrA7*, *aac (3)-1*, CTX-M-1, *bla-Imp*, and *qnrS*) decreased significantly, while other genes (CTX-M-9, *tetM*, *sul1*, and *intI1*) showed no overall decrease (Fig 3B). Two genes (*ermF* and *dfrA12*), showed an increase in absolute abundance after MAD (from $2.3 \times 10^7$ to $2.4 \times 10^8$ copies (g$^{-1}$ dry wt), and from $3.9 \times 10^5$ to $1.4 \times 10^6$ copies (g$^{-1}$ dry wt), respectively). However, the increase in *dfrA12* was not significant.

To gain a more complete understanding of the dynamics of resistance during AD, the relative abundance of 372 genes (including ARGs and MGEs) in each sample was analysed using SmartChip qPCR. Figs 4 and 5 show the results.

Relative abundance of most of the 312 ARGs decreased or fluctuated during TH-MAD or MAD (Fig 4). However, several macrolide and aminoglycoside resistance genes were significantly enriched during both processes. In addition, several tetracycline resistance genes and two glycopeptide resistance genes were enriched specifically during TH-MAD.

The abundance of MGEs dramatically decreased during treatment in both plants (Fig 5). Diversity of all ARG classes and MGEs decreased during treatment, especially during TH, but there was a rebound after AD (Fig 6). In most cases, this rebound did not return ARG diversity to influent sludge levels. Glycopeptide resistance genes were however consistently more diverse in digested sludge than influent sludge. Principal coordinate analysis (PCoA) showed that ARGs were clearly distinguished according to the treatment steps (influent sludge, after TH, after AD) (Fig 7), and that no local or seasonal variation was observed. However, for MGEs, influent sludge clustered together regardless of WWTP, but after AD samples were clearly distinguished according to WWTP.

## Discussion

This study highlights the importance of sewage sludge treatment in the control of ARB and ARG release to the environment. Untreated sewage sludge contains a large number of ARGs, with over a billion copies per gram of sludge for some genes and hundreds of thousands of antibiotic resistant *E. coli* per gram of sludge. This demonstrates that ARB and ARGs are either surviving the aerobic sewage treatment process and/or being diverted from the aqueous phase to sewage sludge in large numbers, and that untreated sewage sludge represents a significant source of ARB and ARGs.

### Fate of antibiotic resistant and virulent *E. coli* during MAD and TH-MAD

Sewage sludge treatment using TH-MAD reduces *E. coli* to undetectable levels, as well as all coliforms and Gram-negatives (S3 Fig). Therefore, the direct risk of resistant potential pathogens being applied to agricultural land is greatly reduced. Although MAD reduces *E. coli* CFUs by more than 99%, they are still present in treated sludge in quantities ranging from $1.2 \times 10^3$ to $4.3 \times 10^3$ g$^{-1}$ dry wt. Overall, 45% of the *E. coli* tested showed resistance to one or more antibiotic, meaning that significant numbers of antibiotic resistant *E. coli* are being added to UK agricultural soil which may pose a direct risk to humans via the food chain, or an indirect risk if these

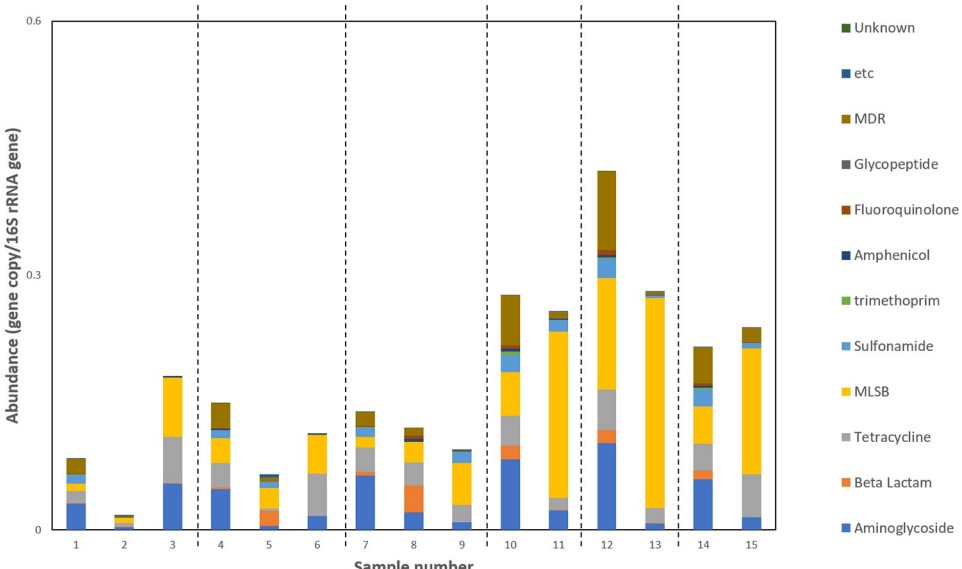

**Fig 4. Relative abundance of ARGs (normalised to 16S rRNA gene) during sludge treatment.** ARGs are grouped according to class. Samples 1–9 are from WWTP 1. Samples 1, 4, 7 are influent sludge, samples 2, 5, 8 are post-TH sludge, samples 3, 6, 9 are post-MAD sludge. Each sample set (1–3, 4–6, 7–9) is from an independent sampling event. Samples 10–15 are from WWTP 2. Samples 10, 12, 14 are influent sludge, samples 11,13,15 are post-MAD sludge. Each sample set (10 and 11, 12 and 13, 14 and 15) is from an independent sampling event.

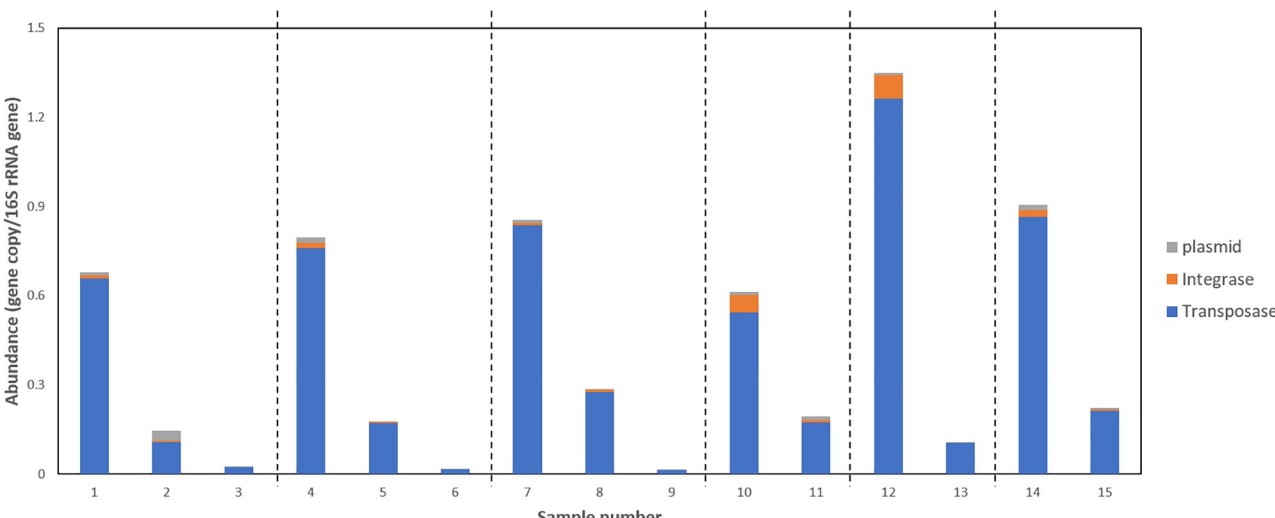

**Fig 5. Relative abundance of MGEs (normalised to 16S rRNA gene) during sludge treatment.** Samples 1–9 are from WWTP 1. Samples 1, 4, 7 are influent sludge, samples 2, 5, 8 are post-TH sludge, samples 3, 6, 9 are post-MAD sludge. Each sample set (1–3, 4–6, 7–9) is from an independent sampling event. Samples 10–15 are from WWTP 2. Samples 10, 12, 14 are influent sludge, samples 11,13,15 are post-MAD sludge. Each sample set (10 and 11, 12 and 13, 14 and 15) is from an independent sampling event.

resistant *E. coli* or their genes spread in the environment. A key future research priority is to determine the fate of ARB and ARGs once sludge is applied to agricultural land.

MAD appears to enrich for trimethoprim resistant *E. coli*. Specific enrichment of certain ARB have been reported previously in the literature [33]. The conditions driving the selection

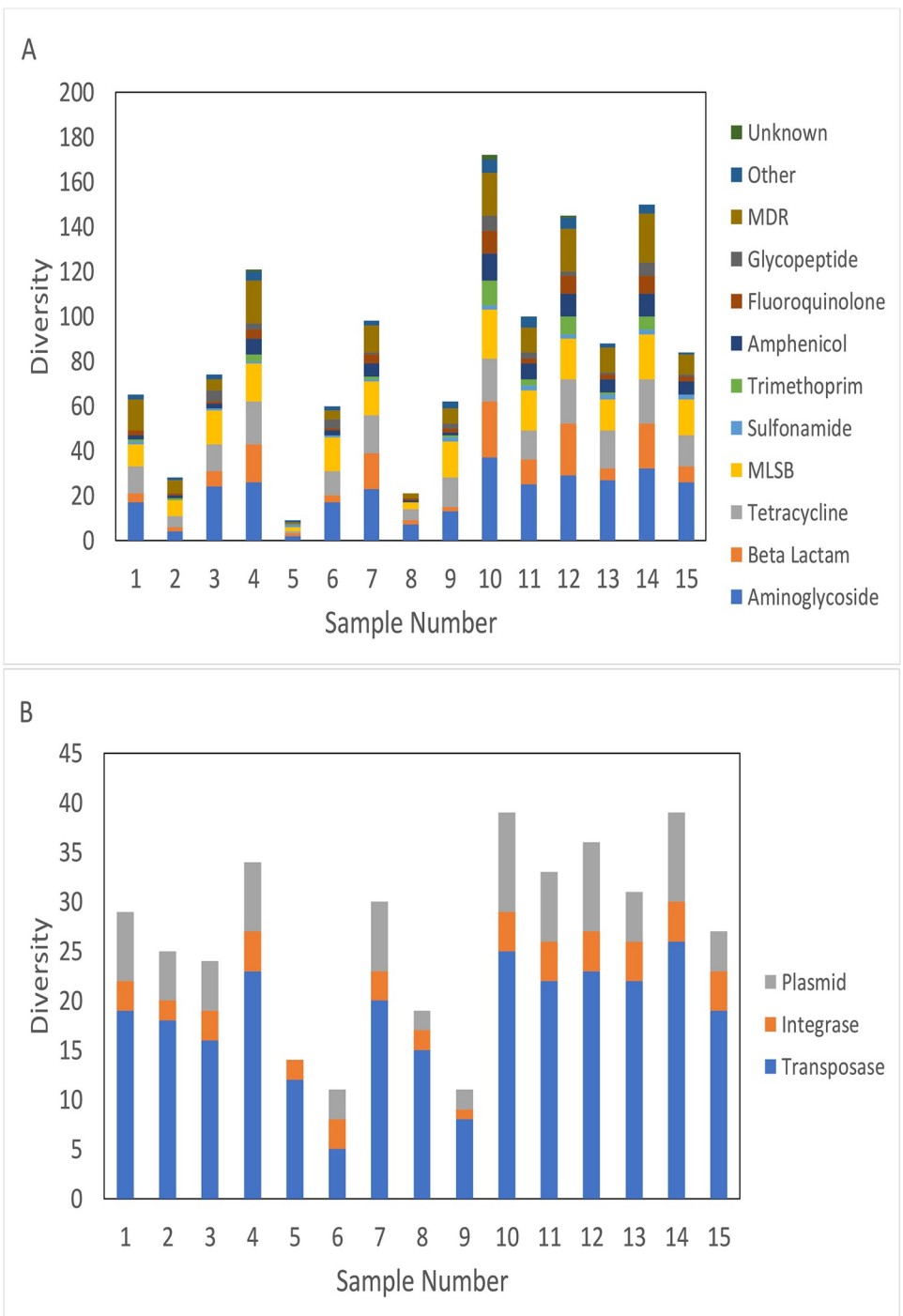

**Fig 6. Diversity of ARGs (A) and MGEs (B) during sludge treatment. Samples 1–9 are from WWTP 1.** Samples 1, 4, 7 are influent sludge, samples 2, 5, 8 are post-TH sludge, samples 3,6,9 are post-MAD sludge. Each sample set (1–3, 4–6, 7–9) is from an independent sampling event. Samples 10–15 are from WWTP 2. Samples 10, 12, 14 are influent sludge, samples 11,13,15 are post-MAD sludge. Each sample set (10 and 11, 12 and 13, 14 and 15) is from an independent sampling event.

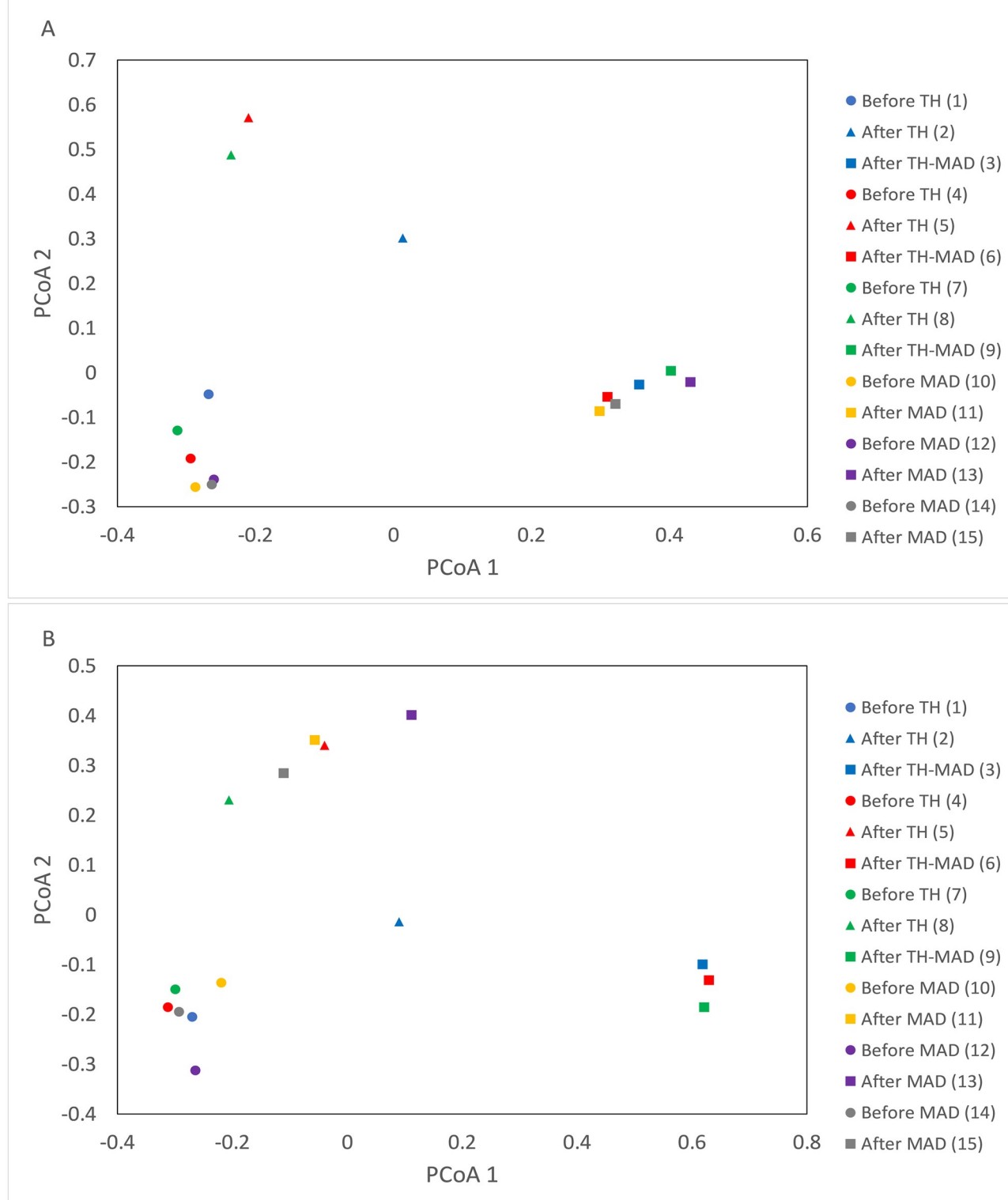

**Fig 7.** **A.** PCoA of ARGs within sludge samples. **B.** PCoA of MGEs within sludge samples. Each colour shows an independent sampling event. Samples taken from the same point in the treatment process are indicated using the same shapes.

of ARB within treatment plants need to be characterised and avoided where possible. No significant change in the abundance of different phylotypes was detected but interestingly, a high prevalence of *E. coli* type B2, responsible for 72% of *E. coli* bloodstream infections [34], was observed.

## Fate of ARGs and MGEs during TH-MAD and MAD

AD decreases the abundance of most ARGs, but macrolide, lincosamide, streptogramin B (MLSB) resistance genes are consistently enriched, in both plants. An increase in MLSB resistance genes has also been reported during anaerobic digestion of sludge in other studies [8, 35]. This apparent enrichment for MLSB resistance genes could be due to carriage in surviving gut bacteria, although this is unlikely as enrichment is also seen after TH, which appears to remove almost all gut bacteria from sludge. MLSB resistance genes could be spreading during AD through horizontal gene transfer, although this seems unlikely due to dramatic decreases in abundance and diversity of MGEs in both plants. MLSB resistance genes could be inherent in a member of the AD microbial community. Interestingly, the only other study to examine the impact of TH on ARGs also found MLSB resistance gene enrichment [25]. One of the enriched genes, *ermF*, showed an increase in absolute abundance during treatment so that treated sludge had a higher abundance than untreated sludge, in both WWTPs. Sequencing of the *ermF* amplicon revealed 100% identity to a plasmid-encoded gene from a clinical isolate (S4 Fig), highlighting the need for further investigation into the fate of these enriched genes in treated sludge. Several aminoglycoside resistance genes are also enriched during AD.

In the TH-MAD plant, tetracycline and glycopeptide resistance genes are also enriched, as are the MGEs *tnpA* (IS6 family) and IS613. IS613 was also enriched during the temperature decreasing stage of composting [36] and has been found to be associated with ARGs [37, 38].

TH reduced the abundance of all genes, but some of these genes rebounded during subsequent MAD. The effectiveness of TH at removing ARGs and antibiotics at pilot scale has previously been reported [39]. Interestingly, an ARG rebound after TH was also observed in that study, which, showed that subsequent thermophilic AD reduced this increase.

The dramatic decrease in diversity and abundance of MGEs during sludge treatment would suggest that anaerobic digesters are not a hotspot for HGT, as has previously been suggested [40, 41]. However, the influence of the small number of MGEs which do increase in abundance; MGEs not included in the assay conducted; or other methods of HGT (transformation and transduction) which were not accounted for in this study cannot be ruled out.

It is reassuring to note that ARG abundance results from both methods of qPCR show similar patterns, providing further evidence for the usefulness of select ARGs such as *sul1*, *tetM* and *ermF* as candidate marker genes to assess overall resistance gene load [27, 42, 43].

PCoA showed that ARGs are clustered according to treatment step with no local or seasonal differences, but MGEs in treated sludge cluster by site. This suggests that conditions within each specific AD plant will influence the abundance and diversity of MGEs, but that this change does not then fundamentally alter ARG abundance and diversity of effluent sludge. However, these differences in the MGE community between different digesters might be responsible for the specific increases in certain genes that were seen and shows that MGEs (and therefore HGT potential) can be very different in different digesters. Tong *et al.* (2019) found that, while MGEs were a strong influence on total ARG abundance, the bacterial and archaeal biomass and community largely influenced the fate of ARGs.

## Conclusions and recommendations

This study is an important step towards filling the research gap on the fate of ARB and ARGs during full scale sludge treatment. The high abundance of ARB and ARGs in raw and treated sludge highlights the importance of sewage sludge as a transmission route for AMR into the environment. ARB and ARGs are applied to agricultural land in vast numbers, which is a potential public health risk.

Measuring the public health risks associated with AMR in the environment is inherently difficult, and many have suggested using the 'precautionary principle' to reduce ARB and ARG release rather than doing nothing. Although AD generally reduces the amount of ARGs and MGEs in sewage sludge, it is a relatively inefficient method for ARG removal. Therefore any 'quick wins' to reduce ARB and ARGs in sludge would be an excellent way of reducing risk, according to the precautionary principle. Thermal hydrolysis is an increasingly common pre-treatment with additional benefits including an increase in methane yield. This research shows that it effectively reduces ARB and most ARGs in sewage sludge, which provides important evidence for the sludge treatment industry considering the installation of new AD plants or the upgrade of existing AD plants. However, the rebound effect shown by some ARGs, and their increase in absolute abundance, deserves further investigation. The increased prevalence of trimethoprim resistant *E. coli* in digested MAD sludge compared to raw sludge is direct evidence of the selection for a resistant potential pathogen within the digestor and may have public health implications when this sludge is applied to agricultural land.

## Supporting information

**S1 Table. Primers and annealing temperatures used in this study.** Table shows primer name and sequence, length (in base pairs) of amplicon, annealing temperature, sources, and (where relevant) which antibiotic the target gene confers resistance to.
(DOCX)

**S2 Table. Reaction matrix used for qPCR of resistance genes, *int1*, and the 16S rRNA gene.**
(DOCX)

**S1 Fig. Comparison of resistance rates in clinical isolates with resistance rates in isolates from WWTP sludge.** The percentage of *E. coli* clinical and WWTP isolates resistant to each antibiotic tested in this study is shown. Clinical data from Public Health Wales (2017).
(DOCX)

**S2 Fig. *E.coli* phylotypes. A**. Before AD, WWTP 2. **B.** After AD, WWTP 2. No significant phylotype shift during AD was observed.
(DOCX)

**S3 Fig. An alignment between the ermF sequence obtained from sewage sludge with a plasmid form of ermF from a clinical isolate of Bacteroides fragilis BLAST sequence ID M14730.1.** There is 100% similarity between the sewage sludge sequence and the plasmid version of ermF.
(DOCX)

**S4 Fig. *E. coli*, other coliforms and non-coliform gram negatives CFUs.** Graph shows before TH, after TH and after TH-MAD at WWTP1, as well as before and after MAD at WWTP2.
(DOCX)

## Acknowledgments

We would like to thank Rhys Jones, Joanne Savvas, Savvas Savvas and Michael Darke of SERC, USW for their assistance in sample collection. We would also like to thank Michael Lott and Lewys Chick for their technical support. Thanks to Paul Henderson for guidance and support with the project. Thanks to our water industry partners for help with sample collection and for providing plant operational data.

## Author Contributions

**Conceptualization:** Sandra Esteves, Richard Dinsdale, Alan Guwy, Emma Hayhurst.

**Formal analysis:** Sky Redhead, Jeroen Nieuwland, Do-Hoon Lee, Dae-Wi Kim, Jordan Mathias, Chang-Jun Cha, Mark Toleman, Emma Hayhurst.

**Funding acquisition:** Sandra Esteves, Emma Hayhurst.

**Investigation:** Sky Redhead, Do-Hoon Lee, Dae-Wi Kim, Jordan Mathias, Chang-Jun Cha.

**Methodology:** Sky Redhead, Jeroen Nieuwland, Do-Hoon Lee, Dae-Wi Kim, Jordan Mathias, Chang-Jun Cha, Mark Toleman, Emma Hayhurst.

**Project administration:** Sandra Esteves, Chang-Jun Cha, Mark Toleman, Emma Hayhurst.

**Resources:** Do-Hoon Lee, Dae-Wi Kim, Chang-Jun Cha, Mark Toleman, Emma Hayhurst.

**Supervision:** Jeroen Nieuwland, Sandra Esteves, Mark Toleman, Richard Dinsdale, Alan Guwy, Emma Hayhurst.

**Validation:** Sky Redhead, Jeroen Nieuwland, Dae-Wi Kim, Chang-Jun Cha, Mark Toleman, Emma Hayhurst.

**Visualization:** Sky Redhead, Jeroen Nieuwland, Do-Hoon Lee, Dae-Wi Kim, Emma Hayhurst.

**Writing – original draft:** Sky Redhead, Dae-Wi Kim, Emma Hayhurst.

**Writing – review & editing:** Jeroen Nieuwland, Sandra Esteves, Chang-Jun Cha, Mark Toleman, Richard Dinsdale, Emma Hayhurst.

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
