## [Decision Letter · Decision Letter 0]

19 Aug 2020

PONE-D-20-22627

An investigation into the fate of antibiotic resistant E. coli and antibiotic resistance genes during full scale conventional and advanced anaerobic digestion of sewage sludge

PLOS ONE

Dear Dr. Emma Hayhurst,

Thank you for submitting your manuscript to PLOS ONE. After careful consideration, we feel that it has merit but does not fully meet PLOS ONE’s publication criteria as it currently stands. Therefore, we invite you to submit a revised version of the manuscript that addresses the points raised during the review process.

As shown below, there are different attitudes on the manuscript. However, the core of both reviews is similar. Please focus the revision on the implications and on the additions.strength this paper delivers compared to the other literature. 

We look forward to receiving your revised manuscript.

Kind regards,

Tim Hülsen

Academic Editor

PLOS ONE

Journal Requirements:

"This work was supported by funds from the Sệr Cymru National Research Network for Low Carbon,

Energy and Environment (NRN-LCEE), from a Welsh Government and European Regional

Development Fund supported under the Smart Expertise Programme SMART CIRCLE project, and by

an internal grant funded by the University of South Wales."

"EH Ser Cymru NRN LCEE Returning Fellowship http://www.nrn-lcee.ac.uk/returning-fellowships.php.en

SE, EH, SR Welsh Government SMART Expertise https://businesswales.gov.wales/expertisewales/support-and-funding-businesses/smart-expertise

We note that one or more of the authors are employed by a commercial company: Independent researcher.

3.1. Please provide an amended Funding Statement declaring this commercial affiliation, as well as a statement regarding the Role of Funders in your study. If the funding organization did not play a role in the study design, data collection and analysis, decision to publish, or preparation of the manuscript and only provided financial support in the form of authors' salaries and/or research materials, please review your statements relating to the author contributions, and ensure you have specifically and accurately indicated the role(s) that these authors had in your study. You can update author roles in the Author Contributions section of the online submission form.

3.2. Please also provide an updated Competing Interests Statement declaring this commercial affiliation along with any other relevant declarations relating to employment, consultancy, patents, products in development, or marketed products, etc.  

Reviewers' comments:

Reviewer's Responses to Questions

**Comments to the Author**

1. Is the manuscript technically sound, and do the data support the conclusions?

Reviewer #1: Yes

Reviewer #2: Yes

2. Has the statistical analysis been performed appropriately and rigorously? 

Reviewer #1: No

Reviewer #2: I Don't Know

3. Have the authors made all data underlying the findings in their manuscript fully available?

Reviewer #1: Yes

Reviewer #2: Yes

4. Is the manuscript presented in an intelligible fashion and written in standard English?

Reviewer #1: Yes

Reviewer #2: Yes

5. Review Comments to the Author

Reviewer #1: Review

The authors present information on the fate of antibiotic resistant E. coli and antibiotic resistance genes in municipal anaerobic digesters with and without thermal hydrolysis pretreatment. The fact that a very large number of ARGs and MGEs were targeted using SmartChip system was interesting to see and a strength of the work. The authors should consider adding more information on the implications of their results and/or future work that may be beneficial based on their results. The following comments are made in a sincere effort to help the authors improve the manuscript.

General comments:

1. I think there are some very practical things that could come out of the results presented. But the manuscript lacks a good, cohesive description of what the results tell us and how they might be used to improve public health and the environment. The authors should add information regarding the implications and usefulness of their results. To describe the usefulness of the results, for example, the authors could also provide a description of future research that would be important to do based on their results. The asuthors should describe in discussion and conclusions sections how the results now help fill the gap described in the introduction.

2. The results lack a description of data precision. Throughout the manuscript, a description of measurement precision is needed along with the range or mean values. What were the detection limits for the genes and E. Coli enumeration? Were values before and after treatment statistically different?

Specific comments:

3.The titles is long and cumbersome. Could “An investigation into the…” be deleted?

4. Line 31, by how much were the 13 ARG abundance values reduced by TH?

5. Line 38, line 198 and throughout the manuscript, was it PCA or PCoA employed?

6. Line 40, I disagree that the results are comprehensive. Two full-scale WWTPs are not a comprehensive sampling of WWTPs.

7. line 78, sewage sludge reuse also poses a risk from other emerging contaminants such as EDCs.

8. line 87, is it an urgent research gap? Describe and justify the urgency. You should describe in discussion and conclusions sections how your results now help fill the gap you describe in the introduction. For example, can your results be used in quantitative risk assessment?

9. Section 2.1, what was the exact temperature and holding time of the digesters? Since pathogen inactivation is a function of these, it is important to know the exact values.

10. line 123, “Sludge is thickened…” which sludge? The mix of primary and secondary? Just the secondary?

11. Line 123, “…SBR)(70%)…” is it percent by volume or mass of solids?

12. Line 126, What is “filter treated effluent?”? Describe.

13. Line 142, why “decanted”? Why take only the supernatant? Does taking the supernatant and not the entire mix influence the results? How?

14. Line 153, briefly describe the CLSI method.

15. line 164, briefly describe the Clermont method.

16. Page 18, there is no section 2.6.

17. Sections 2.5 and 2.7, It is confusing to the reader why two different methods for ARG enumeration were used – SYBR green (QIAGEN) and SmartChip. Please let the reader know up front that two methods were used, why two methods were used, and not just one, why a gene was chosen to be measured by one method and not by another, etc. Why not just use SmartChip?

18. Line 202, “378,000”; and other CFU/g dry weight values in the manuscript. It would be more appropriate to report the log of the results. Also, with three significant figures, you imply your measurements are accurate to +/- 1%. Is this true? What is the standard deviation of the values or log of the values?

19. Line 228, What does “resistance levels” mean? Be more specific.

20, Line 256, change “sample 1 and 3” to “samples 1 and 3.”

21. Line 321, “a high abundance” compared to what?

22. Line 326, “entirely removes”. What was the detection limit of E coli? Is it zero? Can you be sure that it is entirely removed to zero?

23. line 340, in other studies of what? Anaerobic digesters? Activated sludge? Receiving streams?

24. Line 357, “their study”. Who are they?

Reviewer #2: The manuscript, “An investigation into the fate of antibiotic resistant E. coli and antibiotic resistance genes during full scale conventional and advanced anerobic digestion of sewage sludge,” describe work in which the authors examined the fate of resistant bacteria and multiple resistance genes over the length of two treatment trains. This work is highly relevant to global-scale questions regarding antibiotic resistance and WWTPs.

However, I do not feel as if this paper is a good fit for this target journal. As you will see below, I feel that this work doesn’t do enough to separate itself from a wide body of existing literature. PLOS One reports novel and cutting-edge work and, although this is an interesting and well-designed study, it is similar to work that has already been reported.

Specific comments:

Line 23 – a very effective way to start off the abstract is a “big picture” statement. Why is this study relevant in the big picture? The authors do this well in lines 52-56 – I would suggest adding some of the information in lines 52-56 to the beginning of the abstract. This compels the reader to continue reading.

Line 38 – I believe that the authors mean “principal” (rather than principle). I think.

Throughout – the authors use ARBs as an acronym for antibiotic resistant bacteria. This acronym should not be plural.

Line 54 should read, “…as significant emerging environmental pollutants.”

Lines 80-81 – actually, there have been more than 1,500 publications since 2019 in the peer-reviewed literature that focus on sewage sludge (specifically, anaerobic digestion of sewage sludge) and antibiotic resistance. In addition, more than 100 publications have focused on thermal hydrolysis and antibiotic resistance in the same time period. What does your publication lend to this body of literature? You are targeting a high-impact, widely-read journal – it is extremely important here to set your paper apart from this large body of existing literature.

Line 144 – there are literature reports that find low, but measurable false positive rates on MLGA used to identify E. coli. Was any additional testing performed on these isolates to confirm their identity as E. coli? This may have been realized through the phylotyping – but there is no description of what “the Clermont method” is. Perhaps it would be helpful to insert a sentence or two describing this method.

Lines 169-170 – here, you are describing DNA quantity, NOT quality.

6. PLOS authors have the option to publish the peer review history of their article (what does this mean?). If published, this will include your full peer review and any attached files.

Reviewer #1: **Yes: **Daniel H. Zitomer

Reviewer #2: No

---

## [Author Response · Author response to Decision Letter 0]

29 Oct 2020

Response to Reviewers

Reviewer #1: Review

The authors present information on the fate of antibiotic resistant E. coli and antibiotic resistance genes in municipal anaerobic digesters with and without thermal hydrolysis pretreatment. The fact that a very large number of ARGs and MGEs were targeted using SmartChip system was interesting to see and a strength of the work. The authors should consider adding more information on the implications of their results and/or future work that may be beneficial based on their results. The following comments are made in a sincere effort to help the authors improve the manuscript.

Thankyou Reviewer #1 for the positive comments above and we really appreciate the effort to help us improve the manuscript. Our response to each individual point raised is detailed below:

1.I think there are some very practical things that could come out of the results presented. But the manuscript lacks a good, cohesive description of what the results tell us and how they might be used to improve public health and the environment. The authors should add information regarding the implications and usefulness of their results. To describe the usefulness of the results, for example, the authors could also provide a description of future research that would be important to do based on their results. The authors should describe in discussion and conclusions sections how the results now help fill the gap described in the introduction.

We were very aware that we did not want to over interpret our results and were careful to present them without speculation. However, we have now amended our discussion and conclusions section to include potential implications of this research, future research directions and recommendations. We hope this helps give a clearer picture of the importance of our findings.

2. The results lack a description of data precision. Throughout the manuscript, a description of measurement precision is needed along with the range or mean values. What were the detection limits for the genes and E. Coli enumeration? Were values before and after treatment statistically different? 

We have now put mean values and SD into our E.coli values. Our ARG values already show this information. Detection limits for each gene varied for the high throughput qPCR, for single gene qPCR every gene was detected in every sample and therefore we did not reach the detection limit. For E.coli, detection limit was approximately 1 cell per gram.

3.The titles is long and cumbersome. Could “An investigation into the…” be deleted?

Done.

4. Line 31, by how much were the 13 ARG abundance values reduced by TH?

Amounts added (fold change)

5. Line 38, line 198 and throughout the manuscript, was it PCA or PCoA employed?

Apologies – it was PCoA - this has been corrected throughout.

6. Line 40, I disagree that the results are comprehensive. Two full-scale WWTPs are not a comprehensive sampling of WWTPs.

What we meant was that our results are comprehensive within these two specific WWTPs – we have changed the wording to reflect this.

7. line 78, sewage sludge reuse also poses a risk from other emerging contaminants such as EDCs.

We have added in a line about other contaminants.

8. line 87, is it an urgent research gap?Describe and justify the urgency. You should describe in discussion and conclusions sections how your results now help fill the gap you describe in the introduction. For example, can your results be used in quantitative risk assessment?

It is an urgent research gap. Measuring the public health risks associated with AMR in the environment is inherently difficult, and many have suggested using the ‘precautionary principle’ to reduce ARB and ARG release rather than doing nothing until the research quantifies public health risk. We know from other studies that although AD generally reduces the amount of ARGs in sewage sludge, it can be a relatively inefficient method for ARG removal. Therefore any ‘quick wins’ to reduce ARB and ARGs in sludge would be an excellent way of reducing risk, according to the precautionary principle. Thermal hydrolysis is an increasingly common pre-treatment with additional benefits including an increase in methane yield. Investigating its impact on ARB and ARGs provides important evidence for the sludge treatment industry who may be considering the installation of new AD plants or the upgrade of existing AD plants. This has been added to the manuscript. We have also changed the order of our objectives to reflect the importance of this piece of research.

9. Section 2.1, what was the exact temperature and holding time of the digesters? Since pathogen inactivation is a function of these, it is important to know the exact values.

WWTP 2 ranged in temperature from 32.6°C to 38°C over the sampling period. Retention time ranged from 17 to 22 days. Temperature detail, and more specific retention times, have been added to the manuscript. Unfortunately detailed information was not available for WWTP1, however, as stated in the paper we know that the retention time varies between and 12 and 25 days, and we have added in the typical temperature range for mesophlic digestion.

10. line 123, “Sludge is thickened…” which sludge? The mix of primary and secondary? Just the secondary?

The mix of primary and secondary sludge is thickened – this has now been clarified in the manuscript.

11. Line 123, “…SBR)(70%)…” is it percent by volume or mass of solids?

It is by volume – this has been clarified in the manuscript.

12. Line 126, What is “filter treated effluent?”? Describe.

Apologies – this is an error. The plant uses UV treated and filtered final effluent to dilute sludge. This has been amended in the manuscript.

13. Line 142, why “decanted”? Why take only the supernatant? Does taking the supernatant and not the entire mix influence the results? How?

Apologies – ‘decanted’ was the wrong choice of word. The five litre sample was mixed thoroughly to resuspend settled sludge and then 20 ml was taken for further analysis. So each sample was the entire mix, not just the supernatant. This has been amended and clarified in the manuscript.

14. Line 153, briefly describe the CLSI method.

The description in Section 2.3 is the CLSI method so this was already included. We have amended the order of the text to make this clearer.

15. line 164, briefly describe the Clermont method.

This has now been added.

16. Page 18, there is no section 2.6.

Apologies – this has now been fixed.

17. Sections 2.5 and 2.7, It is confusing to the reader why two different methods for ARG enumeration were used – SYBR green (QIAGEN) and SmartChip. Please let the reader know up front that two methods were used, why two methods were used, and not just one, why a gene was chosen to be measured by one method and not by another, etc. Why not just use SmartChip?

Single gene qPCR gives absolute abundance measures, which are important for quantitative risk analysis, and it is a more sensitive technique. High throughput qPCR provides a more comprehensive picture of the fate of ARGs within the sludge treatment plant and does not limit researchers to pre-selected genes. However, it is a less sensitive technique than single gene qPCR and only relative abundance can be quantified, making it less suitable for quantitative risk analysis. Therefore we chose to use a combination of both, selecting 14 genes we felt were representative, commonly found in treatment plants and of clinical interest, but supplementing this with SmartChip analysis which allowed for a much more comprehensive analysis of ARG fate. This explanation has now been added to the manuscript.

18. Line 202, “378,000”; and other CFU/g dry weight values in the manuscript. It would be more appropriate to report the log of the results. Also, with three significant figures, you imply your measurements are accurate to +/- 1%. Is this true? What is the standard deviation of the values or log of the values?

 For consistency we have now changed these to exponential values – we hope this is acceptable. We have also added SD values and changed to 2 significant figures.

19. Line 228, What does “resistance levels” mean? Be more specific. 

We mean the percentage of E.coli isolates resistant to each antibiotic – this has been amended.

20, Line 256, change “sample 1 and 3” to “samples 1 and 3.” 

Done.

21. Line 321, “a high abundance” compared to what? 

This section has been amended to make our point clearer and this term removed.

22. Line 326, “entirely removes”. What was the detection limit of E coli? Is it zero? Can you be sure that it is entirely removed to zero? 

No, we cannot and this phrase has been changed to ‘reduces to undetectable levels’.

23. line 340, in other studies of what? Anaerobic digesters? Activated sludge? Receiving streams? 

MLS genes increase during anaerobic digestion in other studies – this has been clarified.

24. Line 357, “their study”. Who are they? 

Reference 38. This has now been changed to ‘that study’.

Reviewer #2: The manuscript, “An investigation into the fate of antibiotic resistant E. coli and antibiotic resistance genes during full scale conventional and advanced anerobic digestion of sewage sludge,” describe work in which the authors examined the fate of resistant bacteria and multiple resistance genes over the length of two treatment trains. This work is highly relevant to global-scale questions regarding antibiotic resistance and WWTPs.

However, I do not feel as if this paper is a good fit for this target journal. As you will see below, I feel that this work doesn’t do enough to separate itself from a wide body of existing literature. PLOS One reports novel and cutting-edge work and, although this is an interesting and well-designed study, it is similar to work that has already been reported.

Thankyou Reviewer #2 for recognising the global relevance of our research. We feel that our amended manuscript makes clearer how our work is different from existing literature. We have also addressed this comment in more depth under the relevant specific comment below. We believe that this work will be highly relevant to academics, industry representatives and policy makers and will therefore attract a wide readership. For example, in the UK, reports on the topic of AMR and sludge treatment have been commissioned by both public sector (DEFRA) and private sector (UKWIR) organisations, and in both reports the lack of full scale studies was identified as a research gap. By attempting to fill this research gap, our work is novel and cutting-edge. 

Specific comments:

Line 23 – a very effective way to start off the abstract is a “big picture” statement. Why is this study relevant in the big picture? The authors do this well in lines 52-56 – I would suggest adding some of the information in lines 52-56 to the beginning of the abstract. This compels the reader to continue reading. 

This has been added to the abstract.

Line 38 – I believe that the authors mean “principal” (rather than principle). I think

Apologies – changed throughout.

Throughout – the authors use ARBs as an acronym for antibiotic resistant bacteria. This acronym should not be plural. 

Apologies - changed throughout.

Line 54 should read, “…as significant emerging environmental pollutants.” 

Changed as suggested

Lines 80-81 – actually, there have been more than 1,500 publications since 2019 in the peer-reviewed literature that focus on sewage sludge (specifically, anaerobic digestion of sewage sludge) and antibiotic resistance. In addition, more than 100 publications have focused on thermal hydrolysis and antibiotic resistance in the same time period. What does your publication lend to this body of literature? You are targeting a high-impact, widely-read journal – it is extremely important here to set your paper apart from this large body of existing literature. 

We cannot return the numbers of studies quoted by Reviewer #2 when we search using the same keywords. Regardless, there are still very few full scale studies carried out which specifically investigate the fate of ARB and ARGs in during sludge treatment. The vast majority of published works on antibiotic resistance and sludge treatment are run on lab or pilot scale digestors. While these can provide important insights into the behaviour of particular genes or bacteria under idealised conditions with easily controlled variables, they are simply not representative of systems at full scale. To truly understand the effectiveness of sludge treatment then full scale system studies must be carried out. In the few full scale studies that do exist, comprehensive high-throughput qPCR analysis is rare with most studies focusing on just a few ARGs which are not necessarily representative of the whole picture. Few studies also include culture dependent work, but this is a more direct measure of risk which is of relevance to the water industry who are responsible for the safety of sewage sludge. Finally, almost all studies have been carried out in China where treatment processes and climatic conditions are very different to elsewhere. A recent DEFRA-funded report summarised the literature on AMR and AD and highlighted the lack of UK based studies (as an example) as an urgent research gap (Atkins, 2018). Our study gives academic insight into the fate of ARB and ARGs but also gives practical recommendations of relevance to the water industry. This work has been presented at UK water industry events and has received attention and interest. We therefore think that it adds an important piece of work to an under-researched highly complex topic area and is worthy of publication.

Line 144 – there are literature reports that find low, but measurable false positive rates on MLGA used to identify E. coli. Was any additional testing performed on these isolates to confirm their identity as E. coli? This may have been realized through the phylotyping – but there is no description of what “the Clermont method” is. Perhaps it would be helpful to insert a sentence or two describing this method. 

The phylotyping method confirms isolates as E.coli. A description of this method has been added to the paper. We did find a few isolates which were other Escherichia species and these were removed from further analysis – the results presented here are confirmed E.coli isolates.

Lines 169-170 – here, you are describing DNA quantity, NOT quality

Changed as suggested.

---

## [Editor Report · Decision Letter 1]

11 Nov 2020

Fate of antibiotic resistant E. coli and antibiotic resistance genes during full scale conventional and advanced anaerobic digestion of sewage sludge

PONE-D-20-22627R1

Dear Dr. Hayhurst,

We’re pleased to inform you that your manuscript has been judged scientifically suitable for publication and will be formally accepted for publication once it meets all outstanding technical requirements.

Kind regards,

Tim Hülsen

Academic Editor

PLOS ONE
---

## [Editor Report · Acceptance letter]

16 Nov 2020

PONE-D-20-22627R1 

Fate of antibiotic resistant *E. coli* and antibiotic resistance genes during full scale conventional and advanced anaerobic digestion of sewage sludge 

Dear Dr. Hayhurst:

I'm pleased to inform you that your manuscript has been deemed suitable for publication in PLOS ONE. Congratulations! Your manuscript is now with our production department. 

Kind regards, 

on behalf of

Dr. Tim Hülsen 

Academic Editor

PLOS ONE